# Multi-Omics Technologies Applied to Tuberculosis Drug Discovery

**Aaron Goff, Daire Cantillon, Leticia Muraro Wildner and Simon J Waddell \***

Department of Global Health and Infection, Brighton and Sussex Medical School, University of Sussex, Brighton BN1 9PX, UK; A.Goff@bsms.ac.uk (A.G.); D.Cantillon2@bsms.ac.uk (D.C.); L.MuraroWildner@bsms.ac.uk (L.M.W.)

**\*** Correspondence: s.waddell@bsms.ac.uk; Tel.: +44-1273-87-7572

**Abstract:** Multi-omics strategies are indispensable tools in the search for new anti-tuberculosis drugs. Omics methodologies, where the ensemble of a class of biological molecules are measured and evaluated together, enable drug discovery programs to answer two fundamental questions. Firstly, in a discovery biology approach, to find new targets in druggable pathways for target-based investigation, advancing from target to lead compound. Secondly, in a discovery chemistry approach, to identify the mode of action of lead compounds derived from high-throughput screens, progressing from compound to target. The advantage of multi-omics methodologies in both of these settings is that omics approaches are unsupervised and unbiased to a priori hypotheses, making omics useful tools to confirm drug action, reveal new insights into compound activity, and discover new avenues for inquiry. This review summarizes the application of *Mycobacterium tuberculosis* omics technologies to the early stages of tuberculosis antimicrobial drug discovery.

**Keywords:** mycobacterium; tuberculosis; drug discovery; genomics; transcriptomics; proteomics; metabolomics; lipidomics; target identification; mechanism of action; antimicrobial drug resistance (AMR)

## 1. Introduction

Tuberculosis (TB) remains one of the top 10 causes of death worldwide, with 10 million new cases and 1.4 million deaths in 2018. The problem of antimicrobial drug resistance (AMR) is rising, with drug resistance associated with 3.4% of new TB cases globally and up to 50% of previously treated cases in some areas of the world [1]. The discovery of new drugs to treat *Mycobacterium tuberculosis* (*M.tb*) is challenging, with only pretomanid, delamanid and bedaquiline marketed for use in the last 40 years despite sustained international efforts [2]. Multiple logistical and physiological factors contribute to the difficulty of this task (reviewed eloquently elsewhere [3–5]). They include biosafety constraints of working with a slow-growing pathogenic bacterium, heterogeneity of clinical disease and bacterial phenotypes in vivo, intracellular and extracellular *M.tb* sites, drug penetration into lung pathology, the lipid-rich *M.tb* cell wall as a barrier to drug uptake and intrinsic drug resistance, limited number of validated drug targets, the requirement for combination drug therapy, and the length and cost of clinical trials. Omics technologies aim to measure and evaluate together the ensemble of molecular entities by biological class to understand the contribution of each component. Whatever the category of molecule under investigation, the key advantage of omics approaches is that they are unsupervised, and thus less biased by dogma, which is valuable for overcoming drug development bottlenecks [6,7]. Omics in a hypothesis-generating discovery biology setting, is an excellent means of identifying new targets for drug discovery. In a compound-first (discovery chemistry) approach, liberated from reductionist assays, omics technologies are useful tools to reveal or confirm drug mode of action. Mycobacterial omics are

applicable throughout the drug development process from initial drug discovery to preclinical and clinical stages, at each step describing the action of compounds, derivatives, and formulations on *M.tb*. For example, identifying target drift or off-target effects during lead optimization, or characterizing drug resistance conferring mutations in clinical trials.

This review summarizes the application of *M.tb* omics strategies in the early stages of the discovery of new drugs for TB, incorporating genomics (DNA), transcriptomics (mRNA), proteomics (proteins), metabolomics (metabolites) and lipidomics (lipids). The review is not intended to be comprehensive—omics are now fully established in most drug discovery settings—but aims to highlight landmark and interesting approaches to the TB drug development problem. The review centers on omics applied directly to *M.tb*, using examples from other mycobacterial models only to illustrate groundbreaking discovery tools. We focus on (a) target identification, in this context the recognition of potentially druggable pathways worthy of drug discovery efforts in a target-based approach; (b) mode of action studies, often aimed at progressing hits from whole cell compound screening strategies on the long road to the TB clinic (Figure 1).

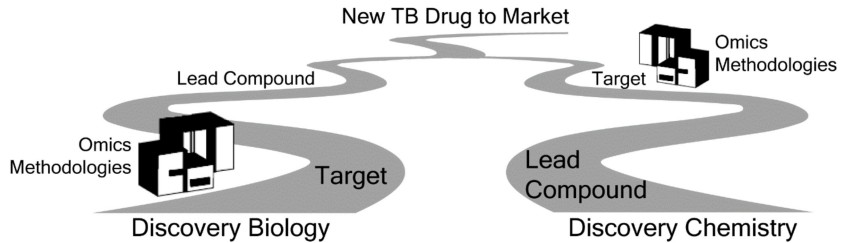

**Figure 1.** The application of omics tools to antimicrobial drug discovery pathways. In a discovery biology approach, omics methodologies are used early in the process to identify new targets for investigation. In a discovery chemistry setting, omics techniques are useful further down the road to identify mechanism of action of lead compounds.

## 2. Genomics

### 2.1. Target Identification

The *M.tb* H37Rv genome was sequenced in 1998, revealing ~4000 potential drug targets, of which ~50% were assigned a tentative function [8,9]. *M.tb* H37Rv was selected for sequencing as the type strain for *M.tb*. *M.tb* H37 was isolated from a patient in 1905 and serially passaged in the laboratory, a resulting strain was named H37Rv, with "R" standing for rough morphology and "*v*" for virulent [8]. It has become a commonly used laboratory strain of pathogenic *M.tb*, a genomic reference for clinical isolates, and a starting point for drug discovery. Since then, comprehensive whole genome sequencing of *M.tb* clinical isolates recovered from patients with TB has uncovered a core genome and mapped the accumulation of single nucleotide polymorphisms (SNPs) in protein coding genes [10]. Comparative genomics has also enabled drug specificity (across microbial species) and toxicity (to mammals) to be anticipated, based on the presence or absence of target protein coding sequences. The application of genomics to *M.tb* has therefore provided a framework of potential drug targets. Manipulation of gene function, through gene inactivation strategies, has aided recognition of pathways that are essential to *M.tb* in different microenvironments, thus highlighting targets for drug discovery programs [6].

Gene deletion methods designed to generate unmarked single gene knockout mutants have been employed alongside global approaches using transposons (Tn) to inactivate gene function [11]. Tn libraries are generated by disrupting genes through random insertion of a transposon throughout the genome, then applying a selective pressure to measure gene essentiality. DNA microarrays were initially used to map the changing abundance of thousands of Tn mutants, transposon site hybridization (TraSH) [12]; more recently coupled to whole genome sequencing (Tn-seq) for greater genomic resolution [13]. Tn mutant screening has been used to map the genetic pathways required for growth of *M.tb* and *Mycobacterium bovis* Bacillus Calmette-Guérin (BCG) in vitro [14]. Here, a *Himar1* based Tn

delivery method using a transducing bacteriophage generated a library of ~100,000 independent clones. A total of 614 genes essential for in vitro growth were found to be evenly distributed throughout the mycobacterial chromosome. In addition, many genes that were shown to be essential for growth appeared to be co-transcribed in operons. Essential genes were identified in amino acid, co-factor and nucleic acid biosynthesis pathways; genes of unknown function were also classified as essential. Many genes assumed to be essential that were predicted to be involved in cell wall and protein metabolism were shown to be dispensable. This is likely due to functional redundancy; for example, *purT* and *purN* both offer alternative pathways for purine de novo biosynthesis and so neither gene was essential [15], providing valuable information that allows non-essential targets to be dropped from drug discovery portfolios. A fundamental pathogenic trait of *M.tb* is the ability to survive and replicate in phagocytes, avoiding phagosome-lysosome fusion and adapting to an intracellular lifestyle [16]. Therefore, to define pathways that are essential for intra-macrophage survival, Barczak et al. mapped genes required for intracellular growth using high content imaging alongside multiplexed cytokine analysis of macrophages infected with *M.tb* Tn mutant libraries [17]. Systematic, multiparametric analysis of *M.tb* Tn mutants impaired for intracellular growth identified functional relationships between *M.tb* Tn mutants and macrophage cytokine profiles. The authors showed that production and export of the complex lipid, phthiocerol dimycocerosate (PDIM), was required for the secretion of ESX-1 substrates and permeabilization of the phagosome, revealing key virulence determinants, alongside defining pathways that are essential to the metabolism of intracellular bacilli, identifying potential targets for drug discovery. Plainly, Tn mutant libraries will only identify genes that are essential in the model under investigation, successful drug targets are likely required to be essential across several different conditions to mimic the variety of microenvironments encountered by *M.tb* during natural infection.

Gene deletion or inactivation results in the absence of gene product, this may not be reflective of drug action, where complete inhibition of protein function may not occur [18]. Conditional expression strategies that reduce rather than abrogate protein function may better represent drug action, and crucially allows essential gene targets to be investigated in the laboratory. Conditional expression using inducible promoter systems (for example, Tet ON/OFF or Pip ON/OFF) allow the expression of essential genes to be increased or reduced to understand gene function, model drug target inhibition and genetically-validate drug targets. The utility of such an inducible gene expression system was highlighted by Johnson et al. where *M.tb* mutants depleted for 474 essential genes (termed hypomorphs) were screened against a large pool of potential inhibitors, allowing for >8.5 million chemical-genetic interactions to take place [19]. An episomally-encoded *sspB* gene was introduced to control protein degradation via a carboxyl-terminal fused DAS-tag. In addition, a 20 nucleotide "barcode" was introduced to facilitate enumeration by sequencing the barcoded PCR products derived from the mutants when pooled. The expression of SspB was controlled by a TetON promoter that was induced in response to anhydrotetracycline; by using TetON promoters with varying strengths, the level of gene product targeted for degradation by the SspB gene could be titrated. Primary screening using this hypomorph methodology identified over 10-fold more hits than whole cell screening wild type *M.tb*. As expected, well known antimicrobial drug classes showed interactions with specific hypomorphs, such as fluoroquinolones with their target GyrA, as well as rifampicin with the β subunit of RNA polymerase. This screening approach identified 39 inhibitors that targeted cellular components that are already clinically-validated antimicrobial drug targets; DNA gyrase, mycolic acid synthesis, and folate biosynthesis, targeted by the fluoroquinolones, isoniazid, and para-amino-salicylic acid, respectively [20]. The 39 compounds were either novel chemical entities or known compounds with re-purposed activity, such as the plant alkaloid tryptanthrin. In addition to finding novel inhibitors for well-validated targets, hypomorph screening also identified inhibitors against novel cellular targets. Johnson et al. demonstrated that strains hypomorphic for the putative efflux pump EfpA were inhibited by the compound BRD-8000 (MIC 6 μM) but this compound showed no activity against wild type *M.tb* (MIC ≥ 50 μM). Subsequent chemical optimization improved activity of BRD-8000 against wild type

*M.tb* to an MIC of 800 nM, an increase of ≥63-fold activity. *M.tb* spontaneous mutants resistant to this modified BRD-8000 compound showed a single point mutation in *efpA*, demonstrating that chemical modification of BRD-8000 had not altered target specificity.

Clustered Regularly Interspaced Short Palindromic Repeats interference (CRISPRi) has the potential to revolutionize the field, enabling precise gene silencing to identify and validate drug targets. A nuclease, dCas9 containing two mutations that eradicate its nuclease activity, is targeted to a mycobacterial gene of interest by a single guide RNA (sgRNA). Upon binding of the dCas9-sgRNA complex to the target site, the DNA duplex destabilizes and prevents gene transcription by blocking RNA polymerase promoter access [21]. Notably, the level of gene silencing can be controlled by varying the sgRNA length and sequence, this allows fine control of the expression of essential genes where traditional gene knockout approaches would be lethal. This system has been further optimized to be induced by doxycycline. This is a lipophilic drug with excellent tissue penetration properties, so that CRISPRi can be employed across a range of drug screening models including in vitro, intracellular and animal studies [22,23]. This approach was used to create libraries containing over 90,000 sgRNAs, generating pools of *M.tb* strains where the majority of genes have been targeted by CRISPRi, enabling high throughput screening approaches to be applied. The utility of CRISPRi in target-based drug discovery was demonstrated through gene silencing of folate metabolism [24]. *M.tb* and mammals require folate; however, *M.tb* must synthesize folate de novo while mammals obtain it through their diet. The variation in folate metabolism makes this biosynthetic pathway an attractive target for antimicrobial drug discovery [25]. While this pathway has been targeted in other bacteria with the antimicrobial drugs trimethoprim, inhibiting dihydrofolate reductase (FolA), and sulfamethoxazole, inhibiting dihydropteroate synthase (FolP1); the action of these drugs in mycobacteria is less clear. Generation of folate biosynthesis knockout mutants in mycobacteria has proved challenging, making target validation difficult for this pathway. *Mycobacterium smegmatis* is a non-pathogenic mycobacterium frequently used as a model organism for *M.tb* due to its low biohazard risk and tractable genomics [26]. Rock et al. utilized a panel of sgRNAs to generate hypomorphs of *folP1*, *folA* and *folC* (dihydrofolate synthase) in this model organism to show that these genes were individually essential. If weaker sgRNAs were used to decrease growth rate rather than inhibit growth completely, there was synergistic growth inhibition, demonstrating the utility of exploiting multiple targets in this pathway to maximize antimicrobial drug activity. Translation of the technology to *M.tb* will yield useful insights into target and pathway essentiality. One caveat to CRISPRi is the potential for off target effects, where dCas9 binds and silences genes that were not intended to be targeted. Bioinformatics packages exist that effectively predict this binding; however, these algorithms may not capture every off-target event [27,28].

## 2.2. Mode of Action

Advances in next generation sequencing technologies and the continued reduction in cost has placed whole genome sequencing (WGS) firmly into the drug discovery pipeline from mechanism of action identification through to monitoring drug resistance post approval [29]. Illumina sequencing (Illumina Inc., San Diego, CA, USA) generates short sequence reads (typically 150 bp long), and while the majority of sequencing data generated today is from Illumina platforms, alternative methodologies are also useful to antimicrobial drug discovery. Long read sequencing (~kilobases), available through platforms such as PacBio (Pacific Biosciences, Menlo Park, CA, USA) and Oxford Nanopore (Oxford Nanopore Technologies Ltd, Oxford, UK), are especially suited for sequencing repetitive regions and structural variations. WGS has been exploited to identify mutations that occur in spontaneous drug resistant colonies growing on solid agar containing a novel antimicrobial compound of interest, typically between 5-fold and 10-fold the minimum inhibitory concentration. Using drug exposure as the selective pressure, this methodology may reveal mutations that affect drug action, in the drug target itself or in gene functions that influence drug activation or efflux. Andries et al. in the discovery of bedaquiline, employed WGS of *M. smegmatis* and *M.tb* bedaquiline-resistant mutants cultured in vitro to demonstrate that this diarylquinoline targets the product of *atpE*; a subunit of the mycobacterial

ATP synthase anchored in the mycobacterial membrane [30]. Kundu et al. further characterized this mode of action, showing that bedaquiline binds to the epsilon subunit [31]. Mutations can, however, be identified in other non-target genes that confer resistance, expanding understanding of drug action but complicating the interpretation of results. For example, in a murine model of TB infection, mutations in the putative Xaa-Pro aminopeptidase *pepQ* were identified that conferred low level resistance to both bedaquiline and clofazimine, neither of which target PepQ directly [32]. While the function of PepQ remains to be determined in *M.tb*, Almeida et al. postulated that this mutation prevented protein degradation of the efflux pump MmpL5, enhancing drug efflux and inducing resistance to bedaquiline and clofazimine [32].

WGS has also been used to reveal the mechanism of action of repurposed licensed pharmaceuticals in *M.tb*. Rybniker et al. identified lansoprazole from the Prestwick library of 1280 FDA approved drugs as protective to lung fibroblasts in an *M.tb* intracellular model of infection [33]. Three lansoprazole resistant mutants were sequenced; each mutant showed the same SNP, a substitution of leucine for proline, in the β-subunit of the cytochrome bc1 complex gene *qcrB*, a key element of the mycobacterial respiratory chain. To demonstrate that exposure to this commonly prescribed proton pump inhibitor did not select for lansoprazole-resistant *M.tb*, Rybniker et al. sequenced 13,559 *M.tb* Complex clinical isolates revealing only one *M. bovis* isolate with a mutation in *qcrB* [34]. Of course, this methodology to define the target of drugs with unknown mechanisms of action is only fruitful if drug-resistant colonies can be raised, which is not always the case. This was highlighted in the discovery of the novel antimicrobial drug, teixobactin from the bacterium *Eleftheria terrae*. Here, a novel iChip culture method was used to capture previously unculturable bacteria in their natural soil environment, which led to the identification of the cell wall-targeting agent teixobactin [35]. Neither *Staphylococcus aureus* nor *M.tb* drug resistant colonies could be generated even at sub-inhibitory concentrations. The authors hypothesized that this indicated general cell toxicity of the compound; however, no mammalian cell toxicity was observed. Subsequent biochemical approaches indicated that this compound exerted its antimicrobial effects through interactions with peptidoglycan pre-cursors—mostly lipid II. Therefore, the inability to generate spontaneous resistance in vitro is likely due to teixobactin inhibiting multiple non-protein targets. Like many other antimicrobial drugs, teixobactin was isolated from an environmental microorganism. It is therefore possible that bacteria in close proximity to *E. terrae* in the soil may be resistant to teixobactin, offering a scenario where teixobactin resistance mechanisms could be discovered [36]. While the inability to detect drug resistance in the laboratory does not mirror the complexity of TB clinical disease alongside host pharmacokinetic and pharmacodynamic factors, the concept of resistance-proof antimicrobial drugs is very attractive and could transform the destructive antimicrobial drug to evolution of drug resistance cycle.

WGS is also shaping our understanding of drug action and drug resistance through the large-scale sequencing of clinical isolates. Consortia mapping drug-resistance conferring mutations have revealed novel mechanisms of resistance and potential novel modes of action of existing anti-*M.tb* drugs in patients [37]. The CRyPTIC Consortium and the 100,000 Genomes Project obtained over 10,000 *M.tb* clinical isolate genomes and associated phenotypes to predict phenotypic drug susceptibility to front line agents from genome sequence. The authors successfully demonstrated that susceptibility could be correlated to the presence or absence of specific antimicrobial drug resistance mutations [37]. This approach also exposed mutations with as yet unexplained roles in drug action. For example, *M.tb* isolates have been identified with mutations in the mycothiol biosynthetic genes *mshA* and *mshC* that confer high-level resistance to ethionamide and low-level resistance to isoniazid [38]. Mycothiol is a key detoxifying and reducing molecule with similar functions to glutathione, which is absent in mycobacteria [39]. Knockout mutants of *mshA* grew normally in vitro and in immunodeficient mice but were growth defective in immunocompetent mice [40], and mycothiol-deficient mutants were less susceptible to ethionamide and isoniazid [41]. Ethionamide and isoniazid are both prodrugs that target the NADH-dependent enoyl-ACP reductase InhA; however, isoniazid is activated by KatG and ethionamide by EthA [42,43]. In order to delineate the roles mycothiol biosynthetic genes play

in isoniazid and ethionamide susceptibility, Xu et al. generated null mutants of each mycothiol biosynthetic gene in *M. smegmatis* [44]. *M. smegmatis* is useful for studying the action of these two drugs, with their target InhA characterized in this bacterium to have >95% sequence identity to the *M.tb* InhA [45]. While the exact role of mycothiol in isoniazid and ethionamide resistance remains to be determined, the authors hypothesized that mycothiol is involved in the activation of isoniazid and ethionamide. *M.tb* KatG, required for isoniazid activation, likely compensates for loss of mycothiol in some settings, resulting in isoniazid susceptibility and ethionamide drug resistance in mycothiol-deficient isolates [40]. Understanding the action of current *M.tb* drugs has direct implications for the development of new inhibitors. For example, bedaquiline and clofazimine resistance can be mediated through SNPs in *Rv0678*, a transcriptional repressor of *mmpL5* and *mmpS5*, which encode an efflux pump of the resistance-nodulation-division (RND) family [46,47]. SNPs in *Rv0678* were detected in clinical isolates of *M.tb* with no prior exposure to bedaquiline or clofazimine [48]; meaning that resistance to bedaquiline was already circulating in *M.tb* isolates before the new drug was introduced to the TB clinic. While the authors could not determine the driver of these mutations, they ruled out rifampicin as a selective pressure. Studies such as these are revealing drug detoxification and efflux pathways that operate across drug class to reduce efficacy of both existing and new antimicrobial drugs.

To expand the methods available for understanding drug mechanism of action, Melief et al. constructed a library of *M.tb* strains overexpressing single genes that could be screened in a high-throughput format [49]. It was hypothesized that library mutants overexpressing the target of an antimicrobial drug should be more resistant than wild type *M.tb* to that particular drug. The library was constructed cloning each gene downstream of a tetracycline-inducible promoter. The 1733 constructs covered 40% of protein coding genes in the *M.tb* genome and contained the majority of annotated essential genes, as well as genes involved in cell wall and fatty acid biosynthesis, virulence factors, regulatory proteins and efflux. The functionality of the system was confirmed by screening the library for resistance to D-cycloserine, which identified the Alr-over-expressing mutant as the only recombinant strain that grew in the presence of the drug. Over-expression of Alr, a target of the drug, resulted in a 7-fold increase in the minimum inhibitory concentration of D-cycloserine. This library represents a new tool to discern targets and pathways that influence drug efficacy of lead compounds with unknown mechanisms of action.

## 3. Transcriptomics

### 3.1. Target Identification

Understanding the mycobacterial transcriptome is key to connecting genome information to protein target expression, highlighting potentially druggable pathways and bacterial responses to drug exposure. With the development of whole-genome technologies, such as microarrays and more recently RNAseq, gene expression studies have been able to capture a snapshot of the total abundance and differential expression of transcripts present in an organism in various conditions. Transcriptomics has become an important tool for exploring the biology of *M.tb*, providing information about adaptive responses to understand mechanisms of pathogenicity, assign gene function, discover new drug targets and explore drug action. Transcriptomics in a discovery biology setting has uncovered induction of potentially druggable pathways involved in β-oxidation of fatty acids, the glyoxylate shunt and cholesterol metabolism in *M.tb* replicating intracellularly in macrophages [50,51] and in expectorated *M.tb* in patient sputa [52], alongside expression of metal detoxification systems [53,54] amongst others. Profiling in vitro models of persistence has revealed adaptations to respiratory and metabolic networks involved in the transition of *M.tb* between different growth states [55,56] highlighting target pathways for investigation. RNA signatures from animal models of TB infection [57,58] and human tissue [59] provide important information on the expression of targets in human disease, enhancing the prospect of cidal drug action by targeting pathways active in vivo. This is valuable evidence for drug discovery decision making, since the bactericidal or bacteriostatic inhibition of an essential target

in vitro does not necessarily predict in vivo drug efficacy. This was demonstrated by Pethe et al. who identified pyrimidine-imidazoles as potent antimycobacterial agents in a whole cell screen against *M.tb*; lead compounds showed activity in vitro but failed to show any inhibition in a murine model of infection [60]. Compound efficacy was linked to the accumulation of glycerol phosphate and a reduction in ATP synthesis in the presence of glycerol. Glycerol metabolism is dispensable in vivo and thus inhibition of this pathway was not cidal in animal models of TB disease. Approaches combining transcriptomic and gene essentiality datasets offer a multi-omics solution to prioritizing pathways for further investigation.

### 3.2. Mode of Action

The transcriptional response of *M.tb* to antimicrobial drug exposure has improved the understanding of many drugs, providing new insights for antibiotics currently in use for TB treatment with known cidal mechanisms, as well as predicting mode of action and the targets of novel compounds. This unsupervised approach is especially useful for understanding the actions of lead compounds from high-throughput screens where the mechanism of *M.tb* killing is entirely mysterious. Comparison of the *M.tb* transcriptional response to a novel compound with mRNA signatures derived from drugs of known function allows broad mode of action to be revealed (Figure 2). In the first study of *M.tb* transcriptional adaptations to drug treatment, Wilson et al. used DNA microarrays to explore changes in gene expression in response to isoniazid [61]. The authors showed that drug exposure induced several genes relevant to isoniazid's known mode of action; drug treatment caused the cluster of five genes encoding type II fatty acid synthase enzymes (*fabD-acpM-kasA-kasB-accD6*) to be upregulated. Other induced genes, such as *efpA* and *ahpC* not in the biosynthetic pathway targeted by isoniazid, were linked to the toxic effects of the drug. In subsequent years, several studies have used DNA microarrays to correlate the mRNA signatures of *M.tb* exposed to antimicrobial drugs with predicted mode of action [62–64]. Boshoff et al. generated a dataset of 430 *M.tb* gene expression profiles to measure the effect of 75 different drugs, drug combinations and growth conditions [64]. The individual RNA drug signatures were classified into groups of agents with similar modes of action (protein synthesis inhibitors, transcriptional inhibitors, cell wall synthesis inhibitors and DNA damaging agents), which have been used to predict the mechanism of action of antimycobacterial compounds of unknown function derived from whole cell screening approaches [65–68].

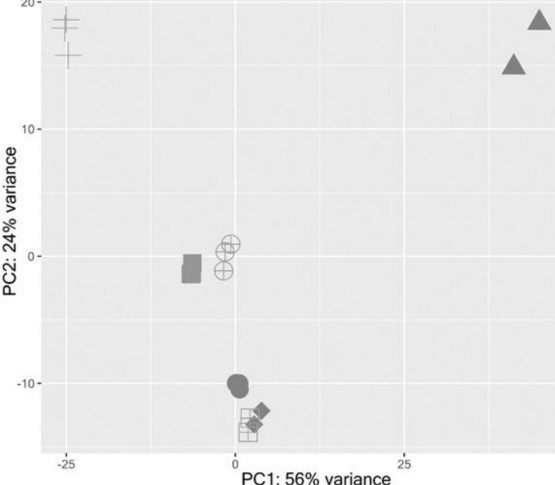

**Figure 2.** The application of transcriptomics to define drug mode of action. Principle Component Analysis (PCA) of *M.tb* responses to seven different drugs (represented by shapes) derived from RNAseq of 2–3 biological replicates per drug. *M.tb* mRNA signatures from antimicrobial drugs with similar mechanisms of action will cluster together.

Transcriptomics has been applied to investigate the mode of action of drugs now in phase I/II studies or in the clinic [2]. The *M.tb* transcriptional signature to benzothiazinone exposure most resembled that of cell wall inhibitors and ethambutol in particular, providing an early indication that the compound was targeting cell wall biosynthesis and arabinogalactan synthesis. This was confirmed to be the case with benzothiazinones inhibiting decaprenyl-phosphoribose-2'-epimerase (DprE1) in the metabolism of D-arabinose, two steps upstream of the action of ethambutol [7]. RNA profiling provided fundamental insights to elucidate the molecular mechanism of mycobacterial killing by pretomanid (formerly PA-824), a nitroimidazole active against both replicating and non-replicating *M.tb*, by inhibiting cell wall synthesis and releasing nitric oxide [69,70]. Pretomanid exposure resulted in a dual signature indicating cell wall inhibition (similar to isoniazid) and respiratory chain disruption (similar to respiratory inhibitors such as cyanide). While the upregulation of genes such as *fasI*, the *fasII* operon, *efpA* and *iniBAC* signposted the aerobic killing mechanism targeting cell wall biosynthesis, drug efficacy in anaerobic conditions was marked by the induction of the *cyd* operon encoding the non-proton-pumping cytochrome bd oxidase, the nitrate reductase *narGHIJ* and other genes involved in respiration [71]. RNAseq analysis of the *M.tb* response to the recently approved nitroimidazole delamanid (formerly OPC-67683) showed many similarities to the pretomanid signature, highlighting that respiratory poisoning plays an important role in the bactericidal effect of these compounds in anaerobic conditions [72].

Boot et al. used RNAseq responses of *M.tb* and *Mycobacterium marinum* to subinhibitory concentrations of ciprofloxacin, ethambutol, isoniazid, streptomycin and rifampicin to select genes to act as drug mode of action-specific reporters [73]. Concordance between the expression levels of *M.tb* and *M. marinum* upon drug exposure were high for the orthologous genes; however, *M. marinum* showed a more distinct stress fingerprint that facilitated simple assays for quick mode of action determination. Ten drug-specific *M. marinum* genes were selected and their promoter regions were cloned into green fluorescent protein (GFP) reporter constructs. As proof of concept that these drug reporters could accelerate TB drug discovery by identifying the mode of action of hit compounds, the *MMAR_4645*-ciprofloxacin reporter and *iniBAC*-isoniazid reporter were used to screen a library of 196 antimycobacterial compounds. The screening revealed two compounds to have a mode of action similar to isoniazid, likely targeting the mycobacterial cell wall, and one compound, similar to ciprofloxacin, that potentially inhibited DNA replication. Understanding mycobacterial responses to antimicrobial drugs may also offer new targets to enhance combination therapy. Peterson et al. used the Environment and Gene Regulatory Influence Network (EGRIN) model and Probabilistic Regulation of Metabolism (PROM) model of *M.tb* regulatory systems to demonstrate that bedaquiline pushed bacilli into a tolerant state that reduced bedaquiline killing [74]. Disruption of this network, by knocking out key transcription factors (*Rv0324* and *Rv0880*) predicted to mediate this response, significantly increased bedaquiline killing. Analysis of transcriptome data from *M.tb* exposed to antitubercular drugs identified molecules that significantly downregulated the expression of *Rv0324* or *Rv0880*, predicting synergism between bedaquiline and pretomanid through the inhibition of the *Rv0880* regulon by pretomanid. The in vitro combination of sub-inhibitory concentrations of both drugs showed an additive to mildly synergistic effect, while the effect was eliminated and a strong antagonism observed when the combinations were tested using a strain over-expressing *Rv0880*. Given the vast number of possible drug combinations, this strategy could complement other preclinical methods and accelerate the discovery of new drug regimens for TB by avoiding combinations with antagonistic interactions and prioritizing those with synergistic effects.

The interaction between drugs in combination were also explored in the computational model INDIGO-MTB, with the premise that drug synergy and antagonism occur due to coordinated, system-level molecular changes involving multiple cellular processes [75]. Using a compendium of publicly-available and in-house *M.tb* transcriptional responses to drug exposure in vitro as input, the model screened in silico more than 1 million potential combinations of 164 drugs, predicting synergistic and antagonistic regimens featuring 35 existing and potential anti-TB drugs. Combinations

containing chlorpromazine, a drug used to treat psychiatric disorders, and verapamil, used to treat hypertension, were highly enriched for synergistic interactions. In contrast, combinations featuring sutezolid, an oxazolidinone anti-TB drug in phase II trials, were observed to be antagonistic. Regimens featuring combinations of bacteriostatic and bactericidal drugs, and combinations of only bactericidal drugs, had significantly more antagonistic interactions than combinations of only bacteriostatic drugs. The predictions of INDIGO-MTB were validated experimentally in vitro using checkerboard assays and the high-throughput DiaMOND method and compared to a meta-analysis of data assembled from 57 phase II clinical trials. The authors found a significant correlation between INDIGO-MTB interaction scores for drug regimen synergy and sputum culture conversion rates after 8 weeks of treatment. The model also identified *Rv1353c* as a key transcriptional regulator mediating multiple drug interactions. The upregulation of *Rv1353c* in vitro reduced drug antagonism of the bedaquiline-streptomycin combination, suggesting this transcriptional factor might be targeted to enhance drug synergies.

Most transcriptional profiling studies map the response of log phase aerobically-respiring bacilli to drug exposure in vitro, this enables direct comparison between drug signatures from axenic culture, but misses the complexity of *M.tb* in vivo phenotypes. Studies by Walter et al. and Honeyborne et al. have characterized drug responses in patient sputa during standard therapy as a measure of in-patient drug efficacy, identifying, for example, an isoniazid signature in expectorated bacilli that disappears after only 3-4 days of the start of drug therapy [76,77].

## 4. Proteomics

### 4.1. Target Identification

Due to limitations in the sensitivity of methods and in the complexity of analysis, proteomics is yet to reach the multifaceted scope of genomics and transcriptomics approaches [78]. However, the variation between protein abundance and corresponding mRNA abundance, suggests that proteomics provides a distinct and useful understanding of *M.tb* physiological responses and target expression [79]. A number of studies have revealed insights into the expression of *M.tb* proteins intracellularly and in vivo, confirming the expression of potentially druggable targets. There are broadly two proteomics strategies, firstly, a top-down approach whereby intact proteins are isolated from a biological sample and sorted by gel electrophoresis, based on their physical and chemical properties, before identification by mass spectrometry (MS) [78]. Popular in early proteomic research, this technique typically found ~100 mycobacterial proteins, accounting for approximately 3% of the total *M.tb* proteome. The second approach is the bottom-up method, where a total set of proteins are proteolytically cleaved into peptides, followed by high-performance liquid chromatography and tandem mass spectrometry (LC-MS/MS). This strategy may quantify many more proteins and track a predefined subset of proteins to greater sensitivity, however computational resolution of profiles and insufficient sensitivity currently limits proteomics discoveries [78]. Despite the lag in proteomics compared to other omics, several studies have revealed new insights into mycobacterial protein expression. Målen et al. compared expression of *M.tb* H37Rv and *M.tb* H37Ra membrane proteins using gel electrophoresis and in-gel digestion of proteins followed by MS [80]. The authors identified over 1700 proteins expressed in both strains, 29 of which were membrane-associated proteins with >5-fold difference in abundance between strains. This highlighted expression of potentially druggable proteins associated with transmembrane transport, complementing genomics and transcriptomics approaches to confirm presence of target protein, and mapped the expression of efflux systems that might impact drug efficacy. Isobaric tagging and stable isotope labelling have enhanced the identification and quantification of proteins by mass spectrometry, enabling more efficient mapping of peptides to protein and allowing comparative samples to be analyzed simultaneously [81]. Thompson et al. quantified N-terminal acetylation of cytosolic and secreted proteins in *M.tb* and *M. marinum* using an N-terminal enrichment approach coupled with stable isotope ratio mass spectrometry [82]. The authors identified 211 endogenously

N-terminally acetylated proteins in *M.tb* (11% of the total observed proteome); 31.8% of these proteins were involved in intermediary metabolism and respiration, indicating the importance and abundance of protein acetylation in mycobacteria. A difference in acylation level was detected in cytoplasmic and secreted forms of 16 mycobacterial proteins, suggesting that N-terminal acetylation might affect protein localization. Acetylation was linked to the secretion of virulence factors (ESX substrates) and antimicrobial drug detoxification systems, signifying N-terminal acetylation as a potentially druggable pathway in *M.tb*. Whilst this approach is clearly beneficial for the detection of acetylation in mycobacteria, consideration should be taken when selecting strains and growth conditions, since these factors will impact protein enrichment resulting in variation between studies, as demonstrated by the authors when comparing the proteome of H37Rv with the published literature.

Proteomics has been applied to in vitro models of infection to find new targets for drug discovery. Albrethsen et al. combined label-free LC-MS/MS and 2D difference gel electrophoresis to reveal extracellular proteins produced by *M.tb* under nutrient starvation [83]. The authors identified proteins involved in toxin-antitoxin (TA) systems, suggesting a role for these systems in the switch from active to non-replicating metabolic states and proposing that TA systems should be explored as potential therapeutic targets. Despite RNAseq often being the preferred approach for such investigations, applying proteomics revealed information regarding protein localization, binding preferences and post-transcriptional modifications that would not be generated from an mRNA analysis. *M.tb* protein expression has been mapped during intracellular infection of alveolar epithelioid cells and macrophages [84]; comparison of the *M.tb* protein expression profiles in these two cell types identified 6 mycobacterial proteins that were differentially expressed in epithelioid cells. Kruh et al. investigated *M.tb* protein expression in vivo, identifying 500 unique *M.tb* proteins that were expressed in the lungs of guinea pigs [85]. Most abundant were *M.tb* proteins implicated in cell wall function and respiration, providing a rational basis for targeted drug discovery against these pathways active during infection.

### 4.2. Mode of Action

Proteomics has helped to deconvolute *M.tb* responses to drug exposure providing insight into mechanisms of drug action and resistance [86]. Recently, Meneguello et al. used proteomics to explore the metabolic pathways that contribute to the activity of rifampicin [87]. A small percentage of rifampicin resistance occurs through mechanisms other than the well-characterized target, β subunit of RNA polymerase. Proteomic profiling of rifampicin-treated *M.tb* resulted in the under-expression of four proteins implicated in cell wall biosynthesis (Ino1, FabD, EsxK and PPE60), suggesting that rifampicin also affects cell wall synthesis contributing to bacterial death. The authors used a liquid chromatography-mass spectrometry (LC-MS) approach, assessing small changes to the *M.tb* proteome temporally. Consideration should be taken when conducting such experiments to minimize the introduction of false-positives as a result of weak signals from an analytical column coupled with a highly sensitive detection platform. Despite these limitations, the authors reported a protein coverage comparable to other studies that applied nano-liquid chromatography. Nano-liquid chromatography is becoming an established tool for advanced peptide separation that uses very narrow columns that are more effective for detecting low abundance compounds [88]. Similarly, proteomics has been utilized to discern the mode of action of repurposed drugs that inhibit *M.tb* in vitro. Sulfamethoxazole, a broad-spectrum antibiotic that primarily targets the folate biosynthesis pathway, exhibits a synergistic effect when combined with other anti-TB drugs. Sarkar et al. mapped the *M.tb* response to sulfamethoxazole exposure using proteomics, identifying induction of oxidative stress and electron transport chain pathways that suggested an additional mode of action for this drug [89]. Proteomics may also be exploited to determine mechanisms of drug resistance. Putim et al. employed a shotgun proteomics system to identify proteins secreted in isoniazid- and rifampicin-resistant *M.tb* compared to drug-sensitive *M.tb*. Bacterial cultures were filtered through low binding protein-cellulose acetate membranes to collect culture filtrate proteins, before sodium dodecyl sulphate-polyacrylamide

gel electrophoresis (SDS-PAGE), in-gel digestion and LC-MS. Depending on the aim of the study, consideration in the liquid chromatography (LC) approach should be taken. For example, for high proteome coverage, or for detecting proteins in low abundance, a 1D gradient for 8 h or a 2D LC should be used. However, for analysis of a limited sample volume, or where characterization of low-abundance proteins is not needed, a 1D gradient for 4 h may suffice. In addition, reproducibility and sample requirements will vary greatly depending on the method used [90]. Differential abundance of proteins involved in lipid metabolism, proteasome function and ATP-binding cassette transporters (ABC transporters) between drug-resistant and drug-sensitive strains may reveal novel systems that influence drug efficacy [91].

## 5. Metabolomics

### 5.1. Target Identification

Metabolomics, the analysis of the metabolite network within a biological system, is an indispensable omics approach for drug discovery, providing information on the potentially druggable processes occurring in a cell. Metabolomics in a discovery biology setting has elucidated *M.tb* metabolic pathways in use in different microenvironments. Carvalho et al. supplemented cultures grown aerobically at 37 °C with different 13C-labelled carbon substrates followed by LC-MS to separate and identify metabolites, demonstrating that *M.tb* co-catabolizes multiple carbon sources simultaneously, through glycolytic, pentose phosphate and tricarboxylic acid pathways [92]. For example, during co-catabolism of dextrose and acetate, dextrose was preferentially metabolized into intermediates of glycolysis and the pentose phosphate pathway, whereas acetate was preferentially used for tricarboxylic acid cycle (TCA cycle) intermediates. This understanding of the *M.tb* metabolic network will help to delineate key pathways and essential metabolites that could be exploited as therapeutic targets. More recently, Serafini et al. used a similar approach to elucidate the metabolic pathways involved in the assimilation of pyruvate and lactate in *M.tb*. Although it is well-established that lipids are important carbon sources for *M.tb* during infection, the authors demonstrated a novel function for the methylcitrate cycle, highlighting that it could be reversed for the biosynthesis of propionyl-CoA and the metabolism of pyruvate and lactate, identifying new targets for drug discovery efforts [93]. This study is an excellent example of a multi-omics approach, combining transposon-directed insertion site sequencing, RNAseq transcriptomics, proteomics and metabolomics, enabling an multi-analyte functional overview of the carbon metabolism network in *M.tb*.

Agapova et al. coupled stable isotope tracing of labelled amino acids with mass spectrometry to elucidate the use of amino acids as a nitrogen source in *M.tb* [94]. The authors showed that the co-metabolism of multiple amino acids as nitrogen sources did not improve growth compared to metabolism of a single source. In addition, several amino acids were utilized as sole nitrogen sources much faster than ammonium, suggesting that *M.tb* preferentially metabolizes specific host amino acids as sources of nitrogen. As such, metabolomics provided insight into the potential for targeting specific pathways in the *M.tb* nitrogen metabolic network. The authors also suggested that greater emphasis should be placed on amino acids as sole carbon sources to better mimic physiologically relevant conditions found in the host. Borah et al. used $^{15}$N-flux spectral ratio analysis to demonstrate that *M.tb* in macrophages has access to multiple amino acids for nitrogen metabolism and identified serine as an amino acid not available to intracellular bacilli [95]. The proteinogenic amino acid serine that provides the nitrogen backbone for glycine and cysteine synthesis must be synthesized by intracellular *M.tb*, highlighting this pathway, and phosphoserine transaminase in particular, as a novel target for drug discovery. In a similar target identification application, Dutta et al. used metabolomics to confirm the role of the stringent response regulator Rel in controlling transition to non-replicating states by comparing wild type *M.tb* with an *M.tb* knockout strain lacking Rel$_{Mtb}$, verifying Rel as a potential anti-TB drug target. The authors then screened a library of compounds against this target, identifying lead compounds that killed nutrient-starved *M.tb* [96].

## 5.2. Mode of Action

The utility of metabolomics is demonstrated in studies to elucidate the mode of action of novel compounds. In a high-throughput metabolomics approach, Zampieri et al. evaluated mass spectra of supernatants from drug-treated *M. smegmatis* cultures to profile a library of 212 antimycobacterial compounds with unknown modes of action [97]. The metabolomic signatures were first established for 62 reference compounds with 17 known targets, before assessing similarity to the test compound profiles. Over 70% of the 212 compounds could be classified with a known mechanism of action, whilst 16 compounds resulted in metabolomic profiles dissimilar to the reference compounds. Of these 16 compounds, 6 exhibited a similar metabolomic response suggestive of the inhibition of lipid and trehalose metabolism. This approach revealed new druggable pathways in *M.tb*, and importantly enables drug discovery programs to diversify target pathways, discarding molecules that likely inhibit targets of existing drugs. In this study, the compounds with unknown mechanisms of action exhibited modest inhibitory activity against *M. smegmatis* with unique metabolic patterns that likely reflect specificity in their underlying modes of action. Whilst further studies should be directed at assessing the activity and mechanism of action of these compounds in *M.tb*, this study clearly demonstrates the utility of this approach to recognize compounds that target novel pathways. In a similar approach, untargeted metabolite profiling using flow infusion electrospray ion high resolution mass spectrometry was used to explore the mode of action of pretomanid [98]. The *M. smegmatis* metabolite profile after exposure to pretomanid was distinct when compared to ampicillin, ethambutol, ethionamide, isoniazid, kanamycin, linezolid, rifampicin and streptomycin-treated cultures. Mapping of differentially abundant metabolites onto pathways highlighted the pentose phosphate pathway, suggesting that accumulation of the toxic metabolite methylglyoxal may contribute to the antibacterial activity of pretomanid. A recent LC-MS-based metabolic linkage analysis of bedaquiline-treated *M.tb* revealed that, alongside inhibition of ATP synthase, glutamine metabolism was also impacted. Since glutamine synthesis inhibitors were synergistic in combination with bedaquiline, an indirect secondary effect of bedaquiline on glutamine biosynthesis was distinguished that could be targeted therapeutically [99]. These approaches demonstrate how metabolomics may be used to elucidate the action of unknown drugs, and reveal fundamental information about the physiology of *M.tb*.

## 6. Lipidomics

### 6.1. Target Identification

Mycobacteria have unique cell envelopes, high in lipid diversity and abundance, comprising up to 40% of the bacillus dry weight [100]. Cell wall biosynthetic pathways are the target of many existing anti-TB drugs; in addition, the sequencing of the *M.tb* genome revealed many lipid biosynthesis and polyketide synthase genes that might be exploited as potential therapeutic targets. The study of this network of cellular lipids within a biological system is broadly categorized in a branch of metabolomics, known as lipidomics, which examines lipid species that are present and how they interact with other lipids, metabolites and proteins in a cell [101]. Lipidomics relies on mass spectroscopy, measuring the mass-to-charge ratio and abundance of gas-phase ions, further characterized into gas chromatography (GC)-MS, liquid chromatography (LC)-MS and direct infusion-MS [102]. The large diversity of lipids and the lack of spatial information about the distribution of these moieties within a cell complicates the inferences from lipidomic experiments [103]. Lipidomics has been employed to discover potentially druggable lipid biosynthesis pathways based on the *M.tb* response to the changing environment. Raghunandanan et al. determined the pattern of *M.tb* lipid changes in hypoxia-induced dormancy and resuscitation, showing that lipid concentration drastically decreased during dormancy and gradually increased again during re-aeration [104]. Several lipids were more abundant in non-replicating bacteria, revealing potentially targetable pathways [104]. This study demonstrates the potential of lipidomics to evaluate *M.tb* in vitro in conditions predicted to mimic in vivo microenvironments. Compared to conventional high performance-LC, the ultra-performance LC technique provided a much

greater chromatographic resolution and subsequently faster analysis time [105]. Lipidomics is also a valuable tool to characterize lipid biosynthetic pathway targets. The fatty acid synthase FAS-II multi-enzyme system is essential for the biosynthesis of mycolic acids and normal cell wall function in mycobacteria. It is the target of several antimycobacterial drugs, such as isoniazid, and therefore offers additional therapeutic potential. To understand the role of HadD, a novel FAS-II enzyme, Lefebvre et al. analyzed total extractable lipids from *M. smegmatis hadD* knockout mutants by high-performance thin-layer chromatography to demonstrate that *hadD* deletion resulted in the absence of α- and epoxy-mycolic acids that disrupted the cell envelope and reduced bacterial fitness [106]. Subsequently, the authors showed that *hadD* deletion in *M.tb* resulted in a 63% reduction in keto-mycolic acids, while overexpression of *hadD* induced an 87% increase in keto-mycolic acids compared to wild type. Knockout mutants of *hadD* were attenuated in a murine model of infection, confirming *hadD* as a new target for drug discovery [107].

*6.2. Mode of Action*

Sharma et al. demonstrated the role for lipidomics in mechanism of action studies by mapping the impact of the natural antimycobacterial compound vanillin in *M. smegmatis*. The authors observed that vanillin changes the composition of fatty acids, glycolipids, glycerophospholipids and saccharolipids causing disruption of cell membrane homeostasis [108]. Similarly, lipidomic analysis has been instrumental in investigating the consequences of *M.tb* drug-resistance conferring mutations to well-characterized anti-TB drugs [109]. Howard et al. demonstrated that rifampicin-resistant *M.tb* isolates with mutations in *rpoB* exhibited altered lipid profiles dependent on the specific location of the SNPs [109]. Lipidomic analysis showed that different *rpoB* SNPs resulted in distinct relative abundances of short-chain and long-chain fatty acid phthiocerol dimycocerosates (PDIM), which induced alternative macrophage activation pathways and altered macrophage metabolism. Such analyses are useful not only to understand the mode of action of a compound, but to deconvolute the consequences of drug resistance-conferring mutations. Lipidomics is often used as an unbiased approach to complement findings from other methodologies. An example is the evaluation of the mycolic acid transporter, MmpL3, as a druggable target [110]. MmpL3 is an integral inner membrane transporter, with a role in the export of mycolic acids to the periplasmic space in the biosynthesis of the mycobacterial cell wall. Lipidomic analysis of *mmpL3 M.tb* knockdown mutants using thin layer chromatography of total lipids revealed a fast decline in cell wall-bound mycolic acids and trehalose dimycolates. Combined with confirmation of in vitro and in vivo essentiality, this study confirmed the mycolic acid transporter MmpL3, as a validated druggable target for *M.tb* drug discovery. Similarly, lipidomics, alongside genomics and transcriptomics, characterized the mode of action of HC2091, a novel compound that likely targets MmpL3. *M.tb* HC2091-resistant mutants were shown to have SNPs in the *mmpL3* gene, likely conferring drug resistance. Thin layer chromatography of the total extractable lipids from HC2091-treated *M.tb* cultures supplemented with radiolabelled-sodium acetate demonstrated a dose-dependent reduction in trehalose dimycolate accumulation with increasing concentration of HC2091. In combination with transcriptional profiling of HC2091-treated bacilli, the authors confirmed that HC2091 targets MmpL3 through a mechanism distinct from other MmpL3 inhibitors [66]. Lipidomics, therefore, often in combination with other omics approaches, is an effective tool, especially in the interrogation of cell wall biosynthetic pathways, a rich source of druggable *M.tb* targets.

## 7. Future Outlook and Conclusions

This review has focused on established mycobacterial omics technologies and their application to the early stages of *M.tb* drug discovery. New omics are emerging that will contribute to future drug development; for example, glycomics, the study of all glycans in a biological system. Glycans are vital in a broad range of processes, their direct recognition by glycan-binding proteins is important for many processes that may be essential and druggable. Recently, Kavunja et al. used a glycomics approach to identify mycolate-interacting proteins associated with synthesis and remodeling of the membrane

in *M. smegmatis* that could lead to the validation of novel therapeutic targets [111]. Such techniques offer unique opportunities for biological discovery and new target identification that will expand as methodologies develop to increase sensitivity and reduce complexity.

The strength of omics technologies is multiplied when used in combination to understand bacterial metabolism and pathogenicity, leading to a true systems approach to antimicrobial drug discovery. This requires the development and maintenance of bioinformatics tools, data repositories, integration and visualization platforms. Recent initiatives in data-sharing not only showcase multi-omics studies but also aim to make omics datasets more readily accessible to the research community [112]. Increased accessibility alongside significant reductions in the cost of omics technologies and advances in data analysis platforms [113], both for data management and for improving ease-of-use, ensure that omics are ubiquitous in drug discovery, preclinical and clinical development programs across academia and pharma [114]. Where *M.tb*-focused omics technologies are combined with human omics systems in a quantitative pharmacology approach to finding and delivering new therapeutics.

The unsupervised nature of omics approaches and the separation from a priori hypotheses places omics technologies in unique and vital roles in the drug development process. For multi-omics to become fully incorporated into the drug development pipeline the challenge remains to move more quickly from initial target identification to target validation in target-based discovery, and in compound-first approaches to scale up mode of action studies to integrate with medium/high-throughput screening. Omics platforms have contributed significantly to the development of the most recent *M.tb* drugs brought to market and to multiple drug candidates now in clinical trials, alongside providing insights into *M.tb* physiology, drug action and drug resistance. Omics methodologies have become valuable tools in the search for new antimicrobial drugs that are becoming increasingly important to find.

**Author Contributions:** Conceptualization; Writing; Review and Editing, A.G., D.C., L.M.W. and S.J.W. All authors have read and agreed to the published version of the manuscript.

**Funding:** This research was funded by the National Centre for the Replacement, Refinement and Reduction of Animals in Research (NC3Rs), grant number NC/R001669/1; and the Wellcome Trust Institutional Strategic Support Fund, project G2306, grant number 192470.

**Conflicts of Interest:** The authors declare no conflict of interest. The funders had no role in the design of the study; in the collection, analyses, or interpretation of data; in the writing of the manuscript, or in the decision to publish the results.

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
