# Peer review of "Multi-Omics Technologies Applied to Tuberculosis Drug Discovery"

_applsci, doi:10.3390/app10134629_

Round 1

Reviewer 1 Report

Multi-omics technologies applied to tuberculosis drug discovery

The authors clearly introduce the very important use of omics technologies in both the discovery biology and the discovery chemistry context, applied to tuberculosis. They review the use of genomics (e.g. identifying which targets are essential and which are dispensable), transcriptomics (e.g. comparing drug actions between different environments, in vitro, in vivo, clinical), proteomics (e.g. actual variation in expression of targets), metabolomics (e.g. contribution of active metabolic processes within the cell to as drug targets), and lipidomics (e.g. lipid varieties in the unique mycobacterial cell envelope as drug target), to both identify targets and reveal mechanism of actions. The authors conclude that the combination of omics technologies are valuable for target identification and mode of action elucidation in a systems approach, which is rightfully envisioned as the future. Experiments using the omics toolbox applied to different species (like Mycobacterium bovis, Mycobacterium smegmatis, Mycobacterium marinum) are reviewed as well, but the authors should reflect on the interpretation of these results in the context of Mtb. How do these species relate to each other? It should also be discussed how the proposed technologies fit within the drug development pipeline regarding both time and costs.

Major comments

  1. In the title only drug discovery phase mentioned. We think that omics techniques have a great value and will become increasingly important for an optimized preclinical and clinical development. Do the authors agree here?
  2. Can the authors briefly mentioned how data from multi-omics can be applied in system biology and quantitative pharmacology?
  3. Is it possible to include a principle figure that you usually obtain in these type of R&D? I think of a component analysis (PCA) of the metabolomics analysis, showing differences in the metabolomes of the different treatment groups (see example below). In addition, a Forest plot of the fold changes of selected lipids from the lipidomics analysis might be added. We think such principle figures would contribute to the broader understanding how data will look like at a stage in their use in drug discovery and development.
  4. The disease area of tuberculosis is only introduced to a minimum extent. Please expand on the need for new drugs and why only three drugs have been approved in the last decades (the bottlenecks mentioned in line 31).
  5. Please shortly introduce the omics methods reviewed, and how they relate to each other, in the introduction. It is also unclear what criteria were used to include methods and papers, please expand.
  6. Please discuss how the results of Mycobacterium smegmatis described in lines 135-140 relate to Mtb. Similarly for lines 201-207, and for the statement in line 454-456.
  7. The interaction between Mtb and macrophage, key in TB pathology, should be introduced in more detail when the relationship between Mtb and macrophages are a hit from genomics (lines 75-82)
  8. It is unclear how the WHO guideline (ref 12) substantiates the claims made in lines 104-106, please review and correct.
  9. Lines 176-185 introduce an interesting point of an agent against which no resistance could be developed. Please reflect what the clinical implications are for future TB treatment.
  10. Please reflect on the impact of the results reported for Mycobacterium marinum in lines 282-290 for Mtb.
  11. It is unclear how the paragraph 305-327 fits within the transcriptomics header, please expand how transcriptomics is input to the computational model.
  12. Similarly, it is unclear why the paragraph 328-340 fits within the transcriptomics header instead of within the genomics header.
  13. Please substantiate claims in lines 25-27, 28-31, 56-58, 355-357, and 504-506 with references.
  14. In conclusion, will or can the authors speculate which of the omics techniques are most suitable for tuberculosis in drug discovery. Is that the same techniques as used in preclinical and clinical development stages?

Minor comments

  1. The introduction introduces the subfield of discovery biology (line 31) but not discovery chemistry (implicitly introduced in line 32-33), please do.
  2. Please be consistent in the use of discovery chemistry (e.g. line 14) vs chemistry discovery (e.g. line 47).
  3. Please emphasize the experimental nature of the H37Rv strain in line 51, in relation to the clinical isolates mentioned in the next line.
  4. Introduce the abbreviation ‘Tn’ in line 65.
  5. Introduce the abbreviation ‘BCG’in line 66.
  6. The readability of the sentence in lines 95-97 will benefit from splitting it into two sentences.
  7. Line 106: ‘…folate, and tryptophan biosynthesis…’
  8. Line 107: ‘In addition to’ instead of ‘In contrast to’
  9. The readability of the sentence in lines 124-127 will benefit from splitting it into two sentences.
  10. Please clarify/emphasize that all three folP1, folA, and folC are essential in line 136-137 (e.g. ‘…and showed that all three of these genes were…’)
  11. Please refer to the company details (location, country) when mentioning the platforms (e.g. Illumina, PacBio, Nanopore)
  12. Line 152: ‘…has been exploited…’
  13. Line 155: should, or could?
  14. Line 184: namely, or mostly? (Lipid II is singular, peptidoglycan precursors is plural)
  15. Lines 205-207: shouldn’t KatG compensation result in isoniazid resistance, instead of susceptibility?
  16. ‘Druggability of targets’ is defined in lines 234-235 in a different way than common in drug development à among others, in vivo drug efficacy (in the text phrased as an addition to druggability) as well as safety, tolerability (and commercial viability) is of importance to druggability. Please correct.
  17. In the conclusion (lines 542-543), an important point on bioinformatics is made, which would benefit from a reference for further consultation.
  18. The abbreviations in lines 378, 405, 407, and 420 are not introduced (but LC is introduced much later in the text in line 480; please introduce it at the first mention).
  19. Please correct the grammar (independent clause?) in line 465-467.
  20. Please correct the grammar (independent clause?) in line 467-469.
  21. Reference list: please review and use consistent format (e.g. capital use in titles, italics for species or genes)
    1. Ref 4 lacks publication year
    2. Refs 5, 33, 45, 49, 88: check journal spelling (U S A)
    3. Ref 8 lacks publication year, page number, DOI
    4. Refs 25, 44, 54, 55, 75, 77, 83, 86 lack page number
    5. Ref 27: check author list (Constortium CR?, the GP?)
    6. Refs 28, 68, 84, 85: confirm page number format
    7. Ref 96 lacks DOI

Author Response

Please see the attachment, thanks.

Reviewer 2 Report

The authors provide a comprehensive review of omics technologies applied to tuberculosis drug discovery.

Following points need to be addressed:

1.  What is the novelty of the review that is presented here? How it is different from the ones already existing. It is essential to highlight these points in the introduction and the abstract

2. I would like to see the duration in terms of years the review has covered the research and review articles that have been previously covered

3. I would like the authors to include an expert opinion/future outlook section before the conclusion and provide perspective

4. Just the mere compliation of studies in paragraphs is not sufficient and the authors should provide the advantages/limitations of the studies that they have covered.

5. In addition, I would also like to see what different approaches/methodologies have been utilized within each of the omics technologies 

6. Are there any studies the authors can include where it shows a path from drug discovery (discovery biology and discovery chemistry) that have led to clinical testing related to tuberculosis drug development

Author Response

Please see the attachment, thanks

Round 2

Reviewer 2 Report

The authors have addressed most of my comments from previous review